

# Risk of phosphorus losses in surface runoff from agricultural land in the Baltic Commune of Puck in the light of assessment performed on the basis of DPS indicator

Stefan Pietrzak[1], Grażyna Pazikowska-Sapota[2], Grażyna Dembska[2], Lidia Anita Dzierzbicka-Glowacka[3], Dominika Juszkowska[1], Zuzanna Majewska[1], Marek Urbaniak[1], Dominika Ostrowska[2], Agnieszka Cichowska[2] and Katarzyna Galer-Tatarowicz[2]

[1] Department of Water Quality, Institute of Technology and Life Sciences in Falenty, Raszyn, Poland
[2] Department of Environmental Protection, Maritime Institute, Gdańsk, Poland
[3] Physical Oceanography Department, Eco-hydrodynamics Laboratory, Institute of Oceanology of the Polish Academy of Sciences, Sopot, Poland

## ABSTRACT

**Background:** In order to counteract the eutrophication of waterways and reservoirs, a basic risk assessment of phosphorus (P) losses in the surface runoff from agricultural land should be included in water management plans. A new method has been developed to assess the risk of P losses by estimating the degree of P saturation (DPS) based on the P concentration of the water extract water-soluble P.

**Methods:** The risk of P losses in surface runoff from agricultural land in the Puck Commune on the Baltic Sea Coast was assessed with the DPS method. The results were compared to an agronomic interpretation of the soil test P concentration (STP). Research was conducted on mineral and organic soils from 50 and 11 separate agricultural plots with a total area of 133.82 and 37.23 ha, respectively. Phosphorus was extracted from the collected samples using distilled water on all soil samples, acid ammonium lactate on mineral soils, and an extract of 0.5 mol HCl·dm$^{-3}$ on organic soils. The organic matter content and pH values were also determined. The results of the P content in the water extracted from the soils were converted into DPS values, which were then classified by appropriate limit intervals.

**Results and discussion:** There was a high risk of P losses from the soil via surface runoff in 96.7% of the agricultural parcels tested (96% of plots with mineral soils and 100% of plots with organic soils). Simultaneously, a large deficiency of plant-available P was found in soils from 62% of agricultural plots. These data indicate that the assessment of P concentration in soils made on the basis of an environmental soil P test conflicts with the assessment made based on STP and create a cognitive dissonance. The risk level of P losses through surface runoff from the analyzed plots as determined by the DPS indicator is uncertain. This uncertainty is increased as the DPS index is not correlated with other significant factors in P runoff losses, such as the type of crop and area inclination.

Corresponding author
Stefan Pietrzak, s.pietrzak@itp.edu.pl

# INTRODUCTION

The Baltic Sea is a basin affected by strong eutrophication (*Andersen et al., 2011*; *HELCOM, 2018b*) resulting in many adverse changes in the marine flora and fauna (*Ojaveer et al., 2010*), which can lead to large social and economic losses (*Ahtiainen et al., 2014*). One of the main reasons for the eutrophication of the Baltic Sea waters is the excessive river inflow of phosphorus (P). The inflow from the entire Baltic basin is estimated at 29.3 tons per year, of which 35.7% comes from dispersed sources, mainly agriculture (*HELCOM, 2018a*). About 12.7 tons of P is estimated to come from Poland to the Baltic Sea by river waters (*HELCOM, 2018a*) and 21–33% of this comes from the agricultural sector, depending on the method of assessment (*National Water Management Board, 2016a*). In order to counteract the eutrophication of the Baltic Sea, various initiatives have been taken at the regional and national level to limit the supply of P to the waters of the Baltic Sea. Important measures were prepared at the Conference of Ministers for the Environment of the HELCOM countries in Copenhagen on 3 October 2013, specifying the expected reduction of the total P loads discharged into the Baltic Sea by the HELCOM member states (*HELCOM, 2013*). Accordingly, Poland is expected to reduce the P inflow to the Baltic Sea by 51% of that expelled during the reference period from 1997 to 2003 (*National Water Management Board, 2016a*). Poland approved this level of reduction as a rough estimate, noting that a final figure would be accepted after relevant analyses were conducted. In light of this measure, Polish agriculture is expected to face major challenges in meeting this requirement, including the impact it will have on inland surface waters. According to Polish monitoring data, eutrophication parameters (nutrient and chlorophyl-a concentration, water transparency) between 2012 and 2015 were exceeded by 42% for sites located on flowing waters (rivers) and by 66% for stands on stagnant waters (lakes/water reservoirs) (*National Water Management Board, 2016b*). Effective solutions to reduce P loss from agricultural sources to waters are needed in light of the Conference's provisions and certain economic factors. Tools and procedures for controlling agricultural lands (AL) in case of P loss in surface runoff should minimize the threat posed by excess P in aquatic ecosystems. The threat to rivers and reservoirs by P loss from agricultural soils has been of interest to researchers for years. This problem is most commonly considered in terms of interactions between the content of plant-available P in soil (determined by various methods, for example, Mehlich 3, Olsen, Egner–Riehm), and P concentration in runoff water; it has been proven that an increase in the P content in soil will correspond to an increased amount of P in surface runoff (*Pote et al., 1996*; *Torbert et al., 2002*; *Sharpley & Kleinman, 2003*; *Brysiewicz, Wesołowski & Pietrzak, 2017*). However, the system for cultivating plants and the type of soil also have an impact (*Gaj, 2008* after: *Sharpley et al., 1981*). The existing agrochemical tests for evaluating P levels are not equally useful for testing in all soil types, limiting their use.

Another applied approach to assessing P loss from agricultural soils via rainwater and the risk of surface water eutrophication is based on the determination of the degree of

P saturation (DPS) (*Alleoni, Fernandes & De Campos, 2014*). The DPS in its classical formula is expressed in a percentage relation of the P content in the soil extract to the P absorption capacity of soil, wherein various approaches are used for the determination of the components in this formula (*Nair et al., 2004*; *Casson et al., 2006*). The procedure for determining the DPS index according to this formula is unique for different types of soil, which can create specific methodological problems and limit the possibilities of its wider use (*Sapek, 2007*). *Pöthig et al. (2010)* recently developed a different method for determining this index, in which the DPS indicator only depends on the P content in soil as determined by the use of distilled water–water-soluble P (WSP). This method is attractive for its simplicity and broad application with different soil types. In the Modelling Nutrient Emissions into River Systems model, developed for quantifying the amount of nutrient emissions from point and diffuse sources in river catchments (*Venohr et al., 2011* after: *Behrendt et al., 2000*), the method proposed by *Pöthig et al. (2010)* is used to determine the P concentration in surface runoff.

This work sought to estimate the risk of P loss in surface runoff from agricultural soils in the Puck Commune using the DPS indicator determined on the basis of WSP, and to identify the practical utility of the DPS index.

## MATERIALS AND METHODS

Research was conducted on the western shore of Puck Bay on the Baltic Sea in the Puck Commune, which is located in the north-eastern region of the Pomeranian Voivodeship, Poland. The Puck Commune is dominated by agricultural land (57.3% of the total area of the commune), the vast majority of which is characterized by high yield potential. The area is largely undulating, with land falls up to approximately 9% (5.14°). Such landform features increase the risk of P loss in surface runoff.

This study is a part of the project "Modelling of the impact of the agricultural holdings and land use structure on the quality of inland and coastal waters of Baltic Sea set up on the examples of the Municipality of Puck region—Integrated info-prediction Web Service WaterPUCK" (*Dzierzbicka-Głowacka et al., 2019*).

In the spring of 2018, soil samples from 61 agricultural plots on 22 farms were taken (field work was conducted with the landowners'/farmers' permission) for chemical analysis based on the guidelines included in the *PN-R-04031 (1997)* standard. An aggregate sample was created by combining and thoroughly mixing a set of primary (individual) samples of up to 20 items. Laboratory analysis of the aggregate samples determined the organic matter content in the layer of 0–5 cm, pH in the layer 0–30 cm, the concentration of plant-available P in layers 0–5 cm and 0–30 cm, and the content of WSP in the layer of 0–5 cm (WSP_w1). In this regard:

1. Fraction content below 0.02 mm was determined by the sedimentation (pipette) method according to the *PN-EN ISO 17892-4 (2016)* standard.
2. Organic matter content was determined as a loss on ignition at 550 °C by the weight method according to the *PN-EN 12879 (2004)* standard.

3. pH measurement of soil was conducted in 1 N suspension of KCl solution by potentiometric method according to *PN-ISO 10390 (1997)* standard (pH$_{KCl}$).

4. The concentration of available P forms (in agronomic soil P test—STP) in mineral soils was determined by acid ammonium lactate (pH ~3.55) according to the *PN-R-04023 (1996)* standard (P$_{AL}$) and by the extract of 0.5 mol HCl·dm$^{-3}$ according to *PN-R-04024 (1997)* standard (P$_{HCl}$) in organic soils; the content of WSP (in environmental soil P test) in mg P·kg$^{-1}$ of soil, was determined by Inductively Coupled Plasma Optical Emission Spectroscopy ICP-OES after drying soil samples in the air and sieving them through a <2 mm sieve, preparing a suspension in the ratio of 1 g of soil in 50 ml of distilled water, agitating for 2 h, and filtering through a 0.45 μm filter.

Soil samples analysis was ordered and evaluated according to the following criteria:

1. Based on the soil organic matter (SOM) content, the soils were divided into mineral and organic soils. With the SOM content of more than 10% (threshold value between mineral and organic soils) organic soil was classified as organic (*Szymanowski, 1995*; *PN-R-04024, 1997*).

2. The assessment of soil acidification was conducted according to the accepted standards. Soils were defined by their reaction classes: very acidic, acidic, slightly acidic, neutral and alkaline under conditions where their measured pH was in the following ranges: ≤4.5, 4.5–5.5, 5.5–6.5, 6.5–7.2 and >7.2, respectively.

3. An assessment of P content in soil was conducted with respect to threshold values given the following standards: *PN-R-04023 (1996)* and *PN-R-04024 (1997)* for the following abundance classes: very low, low, medium, high, and very high (Table 1). The basis for the assessment was the percentage share of the soil samples tested in individual concentration classes.

4. Based on the results of the P content determination in soil using water extract, DPS, in %, indices were calculated using the equation (*Pöthig et al., 2010*):

$$\text{DPS}(\%) = \left\{ 1 / \left[ 1 + \left( 1.25 \cdot \text{WSP}^{-0.75} \right) \right] \right\} \cdot 100$$

where: WSP—is the content of water soluble P, mg P·kg$^{-1}$ of soil.

The risk of P loss by surface runoff was assessed using the limit intervals specified by *Pöthig et al. (2010)* and the established P saturation indexes (Fig. 1). There is a high risk of P loss from the soil by surface runoff if the DPS value exceeds 80%, DPS values lower than 70% were considered safe, and values between 70% and 80% were considered to be tolerable.

The laboratory results were evaluated statistically to determine the basic parameters of the descriptive statistics and any correlations between the analyzed soil indices. The statistics of the results were prepared with Statistica 6 software (*StatSoft, 2019*).

Indoor work and interviews with farmers from 22 farms were conducted to determine the consumption of P fertilizers, the area and type of development of the agricultural plots, the categories, types, soil types and subtypes, and the inclination of the slopes in the research areas. Specially prepared questionnaires were completed by the farmers based on

**Table 1 Soil test P concentrations according to the Polish agronomic interpretation; own elaboration (*PN-R-04023, 1996*; *PN-R-04024, 1997*).**

| Concentration class | Content, mg P·kg⁻¹ of soil dry matter | |
| --- | --- | --- |
| | Mineral soils | Organic soils |
| Very low | ≤22 | ≤174 |
| Low | (22–44) | (174–262) |
| Medium | (44–65) | (262–349) |
| High | (65–87) | (349–523) |
| Very high | >87 | >523 |

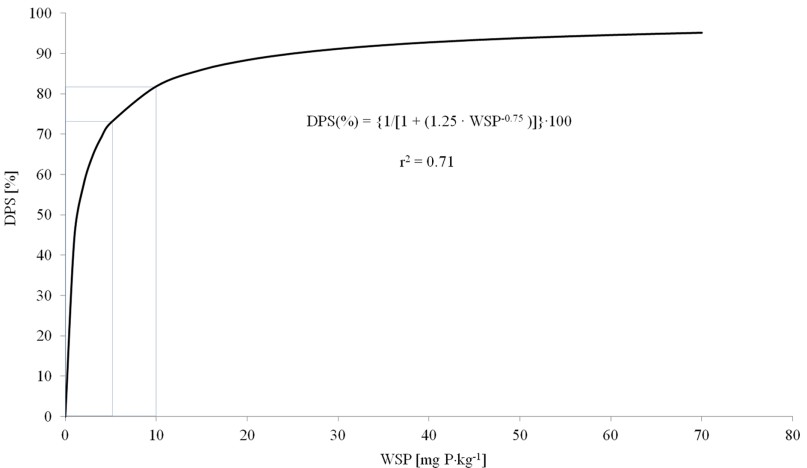

$$DPS(\%) = \{1/[1 + (1.25 \cdot WSP^{-0.75})]\} \cdot 100$$

$$r^2 = 0.71$$

**Figure 1 Correlation between DPS and WSP indicators of soils; own elaboration (*Pöthig et al., 2010*).** $r^2$, value refers to the data of *Pöthig et al. (2010)*.

their own observations of the terrain, a soil agricultural map of the Puck Commune in a vector format (*IUNG-PIB, 1977*), and maps of selected agricultural plots acquired from Google Earth Pro.

## RESULTS

Soils were taken from 61 separate agricultural plots with a total area of 171.05 ha. A total of 50 plots with a total area of 133.82 ha were AL with mineral-derived soils and an organic matter content of 2.53–7.01%; 11 plots were AL with organic soils characterized by organic matter content of 25.60– 68.17% with a total area of 37.23 ha. The following types and subtypes of soils occurred in the research area in the given proportions: brown soils, 61.9%; brown soils, lessive soils, podzolic soils, and rusty soils made from gravel and sands, 10.2%; lessive soils, 3.7%; black soils, 2.4%; and peat and muck–peat soils, 21.8%. Among mineral soils, medium soils with a 21–35% content of particles with diameter less than 0.02 mm were the majority, covering 49.7% of their area, followed by very light (up to 10% content), and light soils (11–20% content), whose share in the mineral soil area was 39.1 and 11.2%, respectively. Grains were the predominant crop on most of the plots with mineral soils, while most of the plots with organic soils were covered with

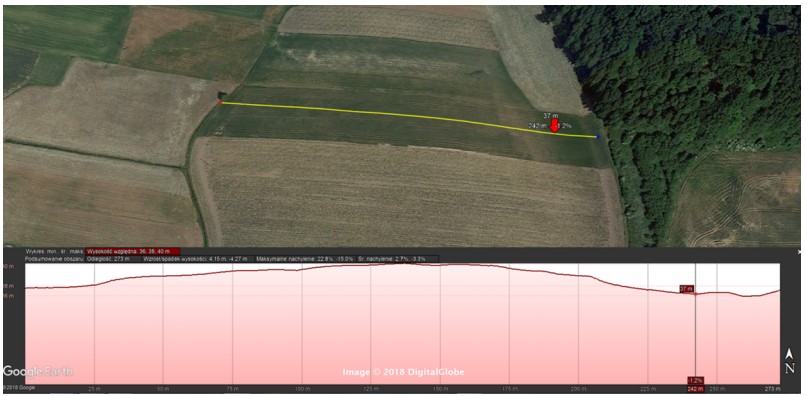

**Figure 2 View of an exemplary agricultural plot from the research area and the shape of its longitudinal profile ((Google Earth Pro 7.3.2.5776: https://www.google.com/earth/versions/). Agricultural plot in Sławutowo (Poland) 54°40′12.58″N, 18°21′1.79″E, elevation 40 M. 3D map. Viewed 22 February 2019).** Map Credit: Google Earth, Digital Globe © 2018.

**Table 2 Distribution of mineral P fertilizers consumption in a group of 22 farms.**

| Specification | P fertilizers doses, kg P·ha$^{-1}$ AL | | | | | |
|---|---|---|---|---|---|---|
| | **0** | **(0–5)** | **(5–10)** | **(10–15)** | **(15–20)** | **>20** |
| Share of farms, % | 13.6 | 9.1 | 22.7 | 31.8 | 4.5 | 18.2 |

permanent grassland. These plots was largely undulating (Fig. 2), with landfalls of up to 9% (5.14°), increasing the danger of P loss from surface runoff.

In 2017, the average consumption of P fertilizers on the farms where the agricultural plots were located was 11.1 kg P·ha$^{-1}$ of AL with a range of 0–24.9 kg P·ha$^{-1}$ AL on the individual farms. This type of fertilizer was used in over 54% of agricultural holdings in doses of 5–10 and 10–15 kg P·ha$^{-1}$ AL (Table 2).

The analyzed soils in the 0–30 cm layer were characterized by a pH$_{KCl}$ within 4.2–7.2 (average 5.4) and 4.9–6.3 (average 5.5) in the case of mineral and organic soils, respectively; this soil depth is treated as a diagnostic layer in the analysis of the agrochemical properties of soils to determine proper fertilizer use. The available P, concentrated in the 0–30 cm layer of the analyzed soils, reached values from 3.6 to 66.5 mg P$_{AL}$·kg$^{-1}$ (average 33.3 mg P$_{AL}$·kg$^{-1}$) in mineral soils, and from 171.0 to 707.0 mg P$_{HCl}$·kg$^{-1}$ (average 340.6 P$_{HCl}$·kg$^{-1}$) in organic soils. The highest amounts of P were found in light soils (average 42.2 mg P$_{AL}$·kg$^{-1}$) while in very light and medium soils the P content was similar (25.6 and 25.8 mg P$_{AL}$·kg$^{-1}$). In the 0–5 cm soil layer the P concentration ranged from 3.6 to 68.8 mg P$_{AL}$·kg$^{-1}$ (average 35.4 mg P$_{AL}$·kg$^{-1}$) in mineral soils and from 136.0 to 526.0 mg P$_{HCl}$·kg$^{-1}$ (average 284.6 P$_{HCl}$·kg$^{-1}$) in organic soils; this soil depth is treated as a standard in environmental research for establishing relationships between the quantitative state of P in soil and surface runoff (Sharpley et al., 1985; Schindler, German & Gelderman, 2002; Schierer, Davis & Zumbrunnen, 2006; Hansen et al., 2012). In this layer, the P concentration in mineral soils was 6.5% higher, on average, than in the

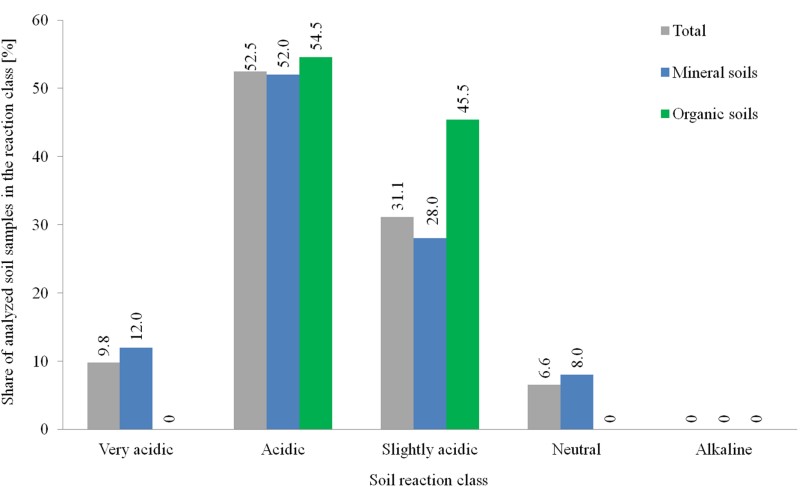

**Figure 3 Distribution of soil pH in tested agricultural plots in the 0–30 cm layer.**

0–30 cm layer. However, the P concentration was 16.6% lower in organic soils. In turn, the WSP_w1 content in the tested mineral and organic soils as determined by water extract was in the range of 2.2–58.5 mg WSP·kg$^{-1}$ (average 24.6 mg WSP·kg$^{-1}$) and 13.7–79.5 mg WSP·kg$^{-1}$ (average 40.3 mg WSP·kg$^{-1}$), respectively. At this level of mineral soils there was 39.0% less P, on average, than in organic soil. In mineral soils the content of WSP_w1 was variable depending on their type; on average it was 20.0, 32.8 and 17.0 mg WSP·kg$^{-1}$ in very light, light, and medium soils, respectively.

## DISCUSSION

### Soil pH and abundance in plant-available P

pH is one of the most important factors determining the physical, chemical, and biological properties of soil. It affects the availability of P accumulated in soil for plants and the P loss to the environment (*Von Tucher, Hörndl & Schmidhalter, 2018*). The soil acidity test revealed that 62.3% of plots were characterized by a very acidic or acidic reaction; 31.1% of plots were slightly acidic and 6.6% were neutral, whereby the share of mineral and organic soils in individual reaction classes was varied (Fig. 3). These conditions are common in the Puck Commune where 13% and 54% of the soils are considered very acidic and acidic, respectively (*Puck Commune Office, 2016*). In the Pomeranian Voivodeship and Poland these fractions reduce to 10% and 31% and 12% and 25%, respectively (*GUS, 2018*).

The range of optimum pH values are assumed to be 5.6–7.0 for mineral soils (in which the majority of cultivated species of plants in Poland grow) (*Kocoń, 2014*) and 4.5–5.5 for organic soils (*Barszczewski, Jankowska-Huflejt & Mendra, 2015* after: *Moraczewski, 1996*). These values represented the soils covering 34.4% of plots, including 30% of plots with mineral soils and 54.5% of plots with organic soils. These data indicate that more than 1/3 of the soil pH from the research areas were very unfavorable from an agronomic and environmental point of view.
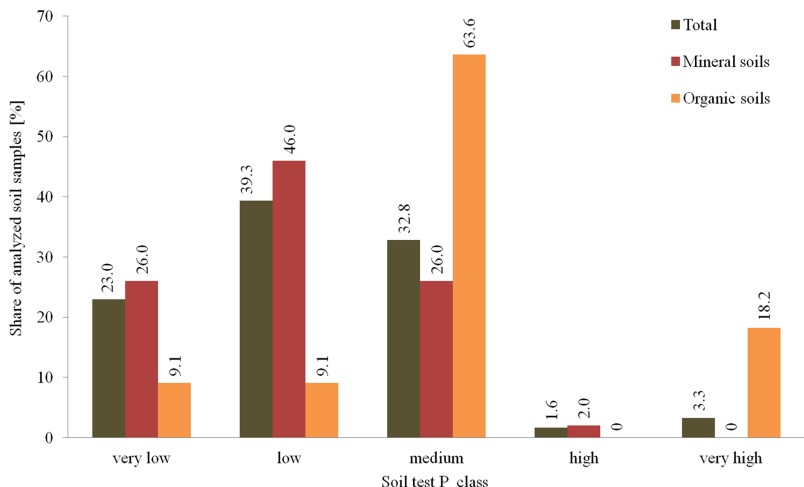

**Figure 4 Distribution of soil P content in tested agricultural plots in the 0–30 cm layer.**

Results of STP revealed that in about 62.3% of plots the P content was very low to low, while 32.8% had a medium content of P (Fig. 4). In 4.9% of all plots, soil fertility had high or very high levels of P. Organic soils were not as abundant in the very low and low classes of P content but were much higher in the medium and very high classes of P content, compared with mineral soils. Soils with high P deficiencies (very low and low P content) should be intensively fertilized with P to protect high plant yields and increase the soil fertility (*Jadczyszyn, Kowalczyk & Lipiński, 2010*). During the research period relatively large P fertilizer doses were used on the farms studied, with an average of more than 11 kg P·ha$^{-1}$ AL. These doses are comparable to the level of P fertilization used on a Poland-wide basis of 10.3 kg P·ha$^{-1}$ AL (*GUS, 2019*). However, these doses exceed the average amount of P fertilizers used in EU countries with well-developed agriculture, such as in Germany where the dose is 7.5 kg P·ha$^{-1}$ AL, or in Denmark and Sweden where the doses are 5.5 kg P·ha$^{-1}$ AL and 3.1 kg P·ha$^{-1}$ AL, respectively (*GUS, 2018*).

The distribution of soil pH and the available P forms concentration had a rather irregular spatial arrangement (Fig. 5). Very low or low P content in the 0–30 cm layer of soil with very acidic and acidic reaction was noted.

## Assessment of DPS indicator

The DPS index was calculated based on the WSP in the analyzed soils and was in the range of 59.1–95.5% (average 88.4%) (Table 3). It should be noted that organic soils were characterized by higher DPS values than mineral soils.

DPS values were set at 80% and higher for soils from 59 plots, including 48 mineral soils and 11 organic soils (Table 4). There was a high risk of P losses to water when the approach to determining this risk suggested by *Pöthig et al. (2010)* was followed, however the accuracy of this method is in question. In order to verify the method, additional studies determining the relationship between WSP/DPS and P concentration in surface runoff need to be conducted. Studies of this nature conducted in northwestern Poland
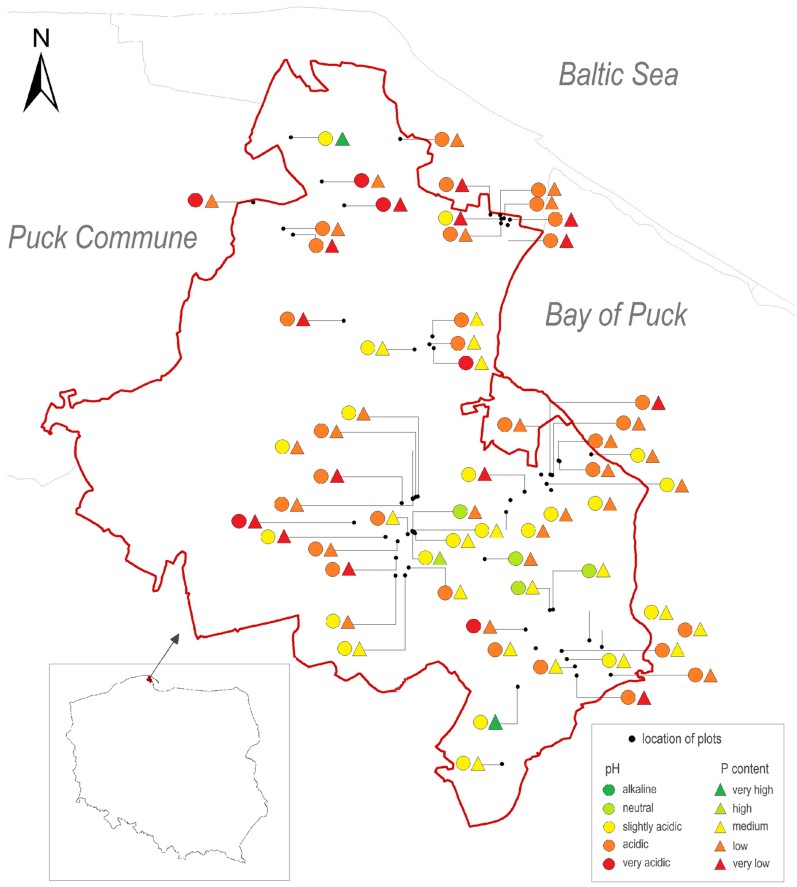

**Figure 5 Spatial arrangement of studied agricultural plots in the Puck Commune with marked soil pH and P content in the 0–30 cm layer; own elaboration.**

**Table 3 Statistical characteristic of the DPS indicator for studied plots.**

| Parameter | DPS value, % | | |
|---|---|---|---|
| | Mineral soils ($n$ = 50) | Organic soils ($n$ = 11) | Total soils ($n$ = 61) |
| $X$ | 87.8 | 91.4 | 88.4 |
| SD | 5.7 | 3.4 | 5.5 |
| Max | 94.4 | 95.5 | 95.5 |
| Min | 59.1 | 85.1 | 59.1 |

Note:
$n$, number of samples; $X$, mean; SD, standard deviation; Max, maximum value; Min, minimum value.

**Table 4 Share of agricultural plots with soils in various DPS classes.**

| DPS classes, % | Risk of P loss from soil | Type of soil | | |
|---|---|---|---|---|
| | | Number of plots (share, %) | | |
| | | Mineral | Organic | Total |
| 70< | lack | 1 (2.0) | 0 (0.0) | 1 (1.6) |
| (70–80) | acceptable | 1 (2.0) | 0 (0.0) | 1 (1.6) |
| ≥80 | big | 48 (96.0) | 11 (100.0) | 59 (96.7) |

Table 5 **Sorption properties of brown soil made of sandy loam which originates from agriculture land located near the research area (54°45′22″N, 18°16′44″E) according to data from 2015 (*Siebielec et al., 2017*).**

| Parameter defining soil sorption properties | Value |
|---|---|
| Hydrolytic acidity (HAC), cmol(+)·kg$^{-1}$ | 4.65 |
| Exchangeable acidity (EA), cmol(+)·kg$^{-1}$ | 0.46 |
| Exchangeable aluminum (ExAl), cmol(+)·kg$^{-1}$ | 0.27 |
| Total exchangeable bases (TEB), cmol(+)·kg$^{-1}$ | 3.99 |
| Cation exchange capacity (CEC), cmol(+)·kg$^{-1}$ | 8.64 |
| Base saturation of soil (BS), % | 46.16 |

(53°16′2.46″N, 14°46′9.42″E) showed that the soluble phosphate concentration (P–PO$_4$) in surface runoff from mineral soils with a P content of 10.3–86.1 mg WSP·kg$^{-1}$ (average 23.2 mg WSP·kg$^{-1}$) ranged from 0.34 to 17.89 mg·dm$^{-3}$ (average 3.11 mg·dm$^{-3}$) (*Pietrzak, Majewska & Wesołowski, 2016*). From mineral soils with an average WSP content of 24.6 mg·kg$^{-1}$ P–PO$_4$ in surface runoff were obtained under specific conditions.

The high DPS values observed over the majority of the research area suggest that P fertilization should be limited for environmental reasons. This conclusion is in contrast to the conclusion made based on the results of the STP; more research is needed to reconcile the observed conflict between the results of P soil tests obtained by agronomic and environmental testing methods. Soil sorption properties researched in close proximity to the current research area as part of the arable soils chemism monitoring in Poland can indirectly assist in understanding these antagonistic assessments (Table 5). The results of this research indicated that the soil was characterized by its content of exchangeable aluminum (ExAl), a relatively low cation exchange capacity (CEC), and low soil buffering capacity. Typical CEC and ExAl values for similar soils in Poland are from 7.18 to 10.38 and from 0.33 to 0.41 cmol(+)·kg$^{-1}$, respectively (*Fotyma & Mercik, 1995*). A small ExAl content in the monitored soil could have been caused by the applied P fertilization which resulted in the reduction of free Al$^{3+}$ ions (*Filipek, 1989*; *Brzeziński & Barszczak, 2009*) due to the precipitation of insoluble aluminum phosphates. In the soils obtained from the majority of research plots, the Al$^{3+}$ content level may have been low, which may explain large deficiencies in available P as established by STP. More reliable assessments of P content in the tested soils were obtained by the agronomic test method compared to environmental test methods.

## Statistical analysis of research results

There were numerous interactions between the analyzed soil indices. These interactions occurred between indices describing mineral soils in which each parameter analyzed was correlated with all others (according to the peer–peer mechanism) (Table 6). These revealed positive, moderate–strong dependencies.

There was a relatively low correlation between the pH$_{KCl}$ value of the soil and the P$_{AL}$ content determined in the layer up to 5 and 30 cm. However, this correlation indicates that by reducing the acidity of soils, the available P would increase. The pH$_{KCl}$ of soil is

**Table 6 Correlations between analyzed indices of mineral soils.**

| Index | Spearman rank correlation coefficient (R) | | | | |
|---|---|---|---|---|---|
| | $pH_{KCl}$ | $P_{AL}\_w1$ | $P_{AL}\_w2$ | WSP_w1 | DPS |
| $pH_{KCl}$ | | 0.3244* | 0.3655** | 0.4017** | 0.4017** |
| $P_{AL}\_w1$ | | | 0.9552** | 0.8474** | 0.7474** |
| $P_{AL}\_w2$ | | | | 0.8176** | 0.8176** |
| WSP_w1 | | | | | 1** |

Notes:
$pH_{KCl}$, soil acidity measured in a suspension of KCl solution; $P_{AL}$, P content in soil determined in acid ammonium lactate; WSP, water soluble P content; DPS, degree of P saturation; w1, soil layer 0–5 cm; w2, soil layer 0–30 cm.
* Correlation significant at $\alpha = 0.05$.
** Correlation significant at $\alpha = 0.01$.

**Table 7 Correlation between analyzed indices of organic soils.**

| Index | Spearman rank correlation coefficient (R) | | | | |
|---|---|---|---|---|---|
| | $pH_{KCl}$ | $P_{HCl}\_w1$ | $P_{HCl}\_w2$ | WSP_w1 | DPS |
| $pH_{KCl}$ | | 0.4771 | 0.4828 | 0.4863 | 0.4863 |
| $P_{HCl}\_w1$ | | | 0.6196* | 0.0866 | 0.0866 |
| $P_{HCl}\_w2$ | | | | 0.3091 | 0.3091 |
| WSP_w1 | | | | | 1** |

Note:
$P_{HCl}$, P content in soil determined in the extract of 0.5 mol HCl·dm$^{-3}$; for additional explanations refer to Table 6.

has a major and direct impact on the P availability. In acidic soils, a large part of this component is immobilized by manganese (Mn), iron (Fe) or aluminum (Al) compounds. The use of liming on such soils increases the amount P available for plant uptake.

The $P_{AL}\_w2$ and $P_{AL}\_w1$ contents were most strongly correlated with each other, which indicated that P was uniformly accumulated in the topsoil (0–30 cm). Therefore, the data regarding the P state may prove equally useful to quantify P loss in surface runoff from agricultural land at each of the mentioned levels. The possibility of using soil samples from levels deeper than 5 cm to assess the risk of P losses in surface runoff was also suggested by *Fischer et al. (2018)* in studies conducted in Brazil that used P determinations in the 0–20 cm layer using the Mehlich-1P test (M1P) to calculate the DPS indicator values on the basis of the M1P–WSP correlation as determined by a set of soil samples.

There were fairly strong correlations between WSP_w1 and $P_{AL}$ content. Similarly strong relationships were found between WSP and $P_{AL}$ content in Brazilian and German soils defined by M1P and calcium-acetate-lactate (CAL) methods based on the P extraction by a mixture of calcium lactate and calcium acetate (*Fischer, 2018*). In case of such correlations, it is possible to convert the results obtained with one method to another with great accuracy. The existence of a relatively strong relationship between WSP_w1 and $P_{AL}\_w2$ and $P_{AL}\_w1$, continued as the latter two indices also remained in a strong relationship with the DPS indicator as a result that DPS is a function of WSP_w1 ($R = 1$).

While all the analyzed indices in mineral soils were correlated, in organic soils only $P_{HCl}\_w1$–$P_{HCl}\_w2$ were statistically correlated, with a moderate degree of correlation (Table 7), ignoring the natural relationship between DPS and WSP_w1.

The correlation between the $P_{HCl}$ content in organic soil at depths of 0–5 and 0–30 cm was weaker than in mineral soils, which may be due to the fact that they were not usually mixed when used and therefore did not favor the homogenization of their top layer composition, including P content. The lack of correlation between WSP_w1, $P_{HCl}$_w2 and $P_{HCl}$_w1 indicates that P transfers from soil to water and to 0.5 mol HCl extracts differently and in disproportionate amounts due to their different extraction possibilities.

## Discrepancy between DPS indicator and STP' results

In the light of the results and analyses, the risk of P loss to waters based on the DPS is arguable. The overall percentage of agricultural plots with soils at high risk for P loss by surface runoff (DPS ≥ 80) was 96.7%, correlating with 96% of plots that had mineral soils and 100% of plots that had organic soils. The obtained results indicate that soils were overly supplied with P and suggest that measures need to be taken in this area to prevent its outflow to waters, for example, by decreasing phosphate fertilizer applications. However, the results of STPs revealed that in order to obtain satisfactory crops, the P content in assessed soils should be significantly increased rather than decreased (therefore, in terms of soil P management various or even contradictory conclusions emerge from environmental and agronomic assessments). Deficient soils were characterized by very low or low levels of P. In the case of plots with mineral soils, 72% were affected by this problem. In cases of serious P deficiencies in the soil, relatively small amounts of this component were found in the surface runoff (*Brysiewicz, Wesołowski & Pietrzak, 2017*). Given the episodic character of runoff, the risk of surface water quality problems under these conditions should not be overestimated and STP and environmental soil P tests may have yielded opposite assessments due to the P retention capacity in deficient soils. Such soils may be marked by low P content as determined by an agronomic test method and a high risk of P release (*Nair & Harris, 2004*; *Nair et al., 2010*).

With regard to the STP, the vast majority of analyzed soils require increased applications of phosphate fertilization and liming in order to optimize their pH for agricultural purposes. This treatment would increase the solubility of the bound form of P in the soil and thus increase the plant-available P. This treatment would be productively justified but could be considered unnecessary in terms of WSP analysis.

Doubts over the adequacy of determining the risk of P loss from soil to water by means of DPS threshold values were increased by the results of the assessment conducted with respect to all analyzed organic soils, which were mainly under grasslands. DPS for these soils exceeded 80% each time so the risk of P loss to waters was high. However, it is difficult to take it as a deciding conclusion as grassland is a biological filter that protects against the release of pollutants into waters. This approach overlooks the contribution of landform features to the uncertain risk of P loss in surface runoff from agricultural soils to waters as determined by the DPS index. In these assessments, the inclination factor should be taken into account as it contributes to the risk of surface runoff (Table 8); significant landfalls which undoubtedly affected the dynamics of P outflow to waters were identified in the research area. The outflow was shaped by a number of other factors,

**Table 8 Assessment of runoff risk for all soil types; own elaboration (*DEFRA, 2005*).**

| Specification | Inclination | | | |
|---|---|---|---|---|
| | >7° (12.3%) | 3–7° (5.2–12.3%) | 2–3° (3.5–5.2%) | <2° (3.5%) |
| Runoff risk | High | Moderate | Lower | Lower |

including the physical and chemical soil properties (pH and organic matter content), soil and plant cultivation methods and atmospheric conditions (*Ulén, 2013*; *Sapek, 2014*). These factors should be considered comprehensively in the identifying of areas susceptible to P losses. This approach is contained in a P index (*Lemunyon & Gilbert, 1993*; *Sharpley et al., 2003*), which is a measure of soil vulnerability to P movement from the field site to waters. The P index is a widely used tool in the USA and Europe to assess the risk of P losses from agricultural landscapes to surface waters (*Buczko & Kuchenbuch, 2007*).

The problem of determining the risk of P loss in surface runoff from agricultural soils to waters is complex. The assessment of such a risk should not depend solely on the value of the DPS index. Therefore, the DPS indicator should be considered in conjunction with a set of parameters characterizing the possibility of P losses from agriculturally used soils to watercourses and water reservoirs.

## CONCLUSION

The research was conducted on a typical undulating area of agricultural land in the Puck Commune, 78.2% of which consisted of arable land and 21.8% of grassland. In terms of mechanical composition, agricultural soils were predominantly medium and light textured, whereas alluvial soils were entirely of organic origin. The majority of the soils were characterized by high acidity (more than 62% of agricultural plots had a very acidic or acid reaction) and a deficiency of plant-available P. In the latter case, over 62% of the analyzed soils, including 72% of mineral soils and over 18% of organic soils, were characterized by very low and low P content. The DPS exceeded 80% in almost all of the assessed soils as determined on the basis of P content extracted in the water extract. There was a high risk of P loss in surface runoff to water after taking into the account the existing criteria of the DPS index assessment. However, this assessment should be approached cautiously as it does not correspond to the results of the agronomic assessment of the P content in soil in terms of the requirements for the application of phosphate fertilization. It also disregards other factors affecting P loss in runoff, such as the type of crop or area inclination.

The research results show that the risk assessments of P losses from agricultural soils performed on the basis of soil P content testing with an environmental test method (e.g., using WSP content and calculated DPS indicator) and an agronomic one (e.g., using P–AL content) can be antagonistic. This indicates that there is still a problem in determining the risk of P loss in surface runoff from agricultural land to water and further research is needed to develop a solution.

### Funding

This work was supported by the National Centre for Research and Development within the BIOSTRATEG III program No. BIOSTRATEG3/343927/3/NCBR/2017. The funders had no role in study design, data collection and analysis, decision to publish, or preparation of the manuscript.

### Grant Disclosures

The following grant information was disclosed by the authors:
National Centre for Research and Development within the BIOSTRATEG III program:
BIOSTRATEG3/343927/3/NCBR/2017.

### Competing Interests

The authors declare that they have no competing interests.

### Author Contributions

- Stefan Pietrzak conceived and designed the experiments, analyzed the data, prepared figures and/or tables, authored or reviewed drafts of the paper, and approved the final draft.
- Grażyna Pazikowska-Sapota conceived and designed the experiments, performed the experiments, analyzed the data, authored or reviewed drafts of the paper, and approved the final draft.
- Grażyna Dembska performed the experiments, analyzed the data, prepared figures and/or tables, and approved the final draft.
- Lidia Anita Dzierzbicka-Glowacka conceived and designed the experiments, analyzed the data, authored or reviewed drafts of the paper, and approved the final draft.
- Dominika Juszkowska analyzed the data, prepared figures and/or tables, and approved the final draft.
- Zuzanna Majewska analyzed the data, prepared figures and/or tables, and approved the final draft.
- Marek Urbaniak analyzed the data, prepared figures and/or tables, and approved the final draft.
- Dominika Ostrowska performed the experiments, analyzed the data, prepared figures and/or tables, and approved the final draft.
- Agnieszka Cichowska performed the experiments, analyzed the data, prepared figures and/or tables, and approved the final draft.
- Katarzyna Galer-Tatarowicz performed the experiments, analyzed the data, prepared figures and/or tables, and approved the final draft.

### Field Study Permissions

The following information was supplied relating to field study approvals (i.e., approving body and any reference numbers):
Fieldwork was conducted with the landowners'/farmers' permission.

## Data Availability

The raw data is available as a Supplemental File.

## Supplemental Information

Supplemental information for this article can be found online at http://dx.doi.org/10.7717/peerj.8396#supplemental-information.

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
