# Peer review of "Risk of phosphorus losses in surface runoff from agricultural land in the Baltic Commune of Puck in the light of assessment performed on the basis of DPS indicator"

_PeerJ, doi:10.7717/peerj.8396_

## Round 0.1 · original submission · Major Revisions

Both reviewers had extensive comments on your manuscript which need to be addressed in your revision. In addition, once you have made all the necessary revisions, I strongly suggest that you have the revised manuscript edited for grammar. I will not accept a revised manuscript that is not readable.

Reviewer 1 ·

Basic reporting

The manuscript would clearly benefit language edition for correctness and clarity.
Argumentation would be stronger if more references would be used to discuss the conflict between environmental and agronomic interpretations.
All raw data is not shared.
My impression is that the current data is not enough to show which assessment is more correct.
Comments are given in General Comments section.

Experimental design

This is an original research.
The question is clearly defined, but I have some doubt if it can be answered with the presented data. However, this is an important topic which I wish the authors can elaborate further.
Some details on methods should be added.
Comments are given in General Comments section.

Validity of the findings

All data are not presented.
The authors should work on to provide direct evidence on their conclusions.
Comments are given in General Comments section.

Additional comments

Review of PeerJ manuscript “Risk of phosphorus loss in surface runoff from agricultural land in the Baltic Commune of Puck” by Pietrzak et al.
The manuscript describes a soil survey study in the Polish Baltic Sea coast, with an aim to test a P loss risk evaluation method. The manuscript is based on soil samples that are analyzed using various soil P tests. This manuscript deals with an important topic when Baltic Sea countries try to curb P losses from land to sea, and agriculture is on the central position because of the high share of anthropogenic P loading to the Sea originating from agricultural activities.
The data shows an interesting conflict between environmental and agronomic assessments. Which of the two assessments is more biased remains an open question with the data provided. A major issue is that there are no direct measurements on P losses, or data that would show that soil P status would restrict plant growth. However, this conflict is interesting by itself and should be studied further, because plants are efficient in extracting P from very low concentrations of soil solution. If the soils release high concentrations of P in water extraction, it is puzzling how the plants would be deficient in P. In its present form the data does not provide any answer on this discrepancy. I especially missed data on reactive Al and Fe contents, because they are the main P sorption components in non-calcareous soils and their concentration may vary widely in different geographic areas. These data would help in understanding P reactions in the studied soils.
As the authors mention, degree of P saturation (DPS) it is traditionally derived from as measured soil P content as related to the major P sorbing constituents (extractable Al and Fe). Because DPS is an established term for this kind of an approach, it should not be used for other ones. It is unfortunate that the term has been already used by some authors, because it will create confusion in readers who are not very familiar with environmental soil science. The equation of Pöthig (2010) is merely a model that uses soil WPS concentration to estimate the relationship between P sorption maximum and a soil’s total P content. The equation is not likely a “universal” one, because not all P associations are similarly contributing to WSP (e.g. native apatite-P is a very stable P compound), which affects the ratio of P sorption maximum to total P. It may be that the data presented here is an indication of non-universality of the approach.
The manuscript would benefit from editing the English by a native person who is familiar with soil science. I also suggest that you use throughout the text more specific terms and expressions (see the detailed comments below). One such terminology issue is “available P” and “P content”, because the former can be only obtained with exhaustion experiments with plants and the latter rather refers to total P stock. Use instead the term “agronomic soil test P (STP)”.
My suggestion is that the manuscript is returned to the authors for complementation and corrections. It is necessary that conflicts between environmental and agronomic assessments are studied and published. However, one should be able to show with measured data which one has a flaw in it. Unfortunately the present data is not sufficient to clarify this.
Detailed comments:
23: The word “determination” implies that DPS is actually determined by analyzing sorbed P and the sorption components (Al, Fe), but this approach is rather “estimation based on P concentration of water extract”.
26: You could be more specific here: “a critical analysis of assessment results was carried out”; I think it’s more like “compared to the agronomic interpretation of soil test P concentrations”.
29: “P content was determined” could be changed to “P was extracted”…
35: “ascertained” is a rather strong word, because deficiencies were not quantified. The agronomic interpretation on what P concentrations are considered sufficient also changes with time, see e.g. Buczko et al. 2017 (DOI 10.1007/s13280-017-0971-1), because large safety margins were commonly applied in the past when making P recommendations and in setting limits for optimal soil test P.
36-39: I would like to see a more decisive statement at the end of Abstract. There is clearly a conflict between the environmental and agronomic interpretations, or on how the environmental and agronomic soil tests work on these soils.
42-72: It would be good to back the statements in the start of Introduction with references. Now there’s only two sources used, Helcom and National…(could the National Water Management Board be referred in text to as NWMB?). Research on these issues should be given a credit, not only reports or summaries that are based on them.
Please avoid unclear references to what has been said previously, because they may confuse a reader. As example line 51 (of the mentioned component), 61 (the above), 67 (the component in question), … It is better to be specific so that a reader does not to need go back to the previous sentence and look what these words refer to.
92: “P saturation in surface runoff”, change to “P concentration in surface runoff”.
93-98: This paragraph would benefit from rewriting. For testing the approach’s usefulness in predicting P loss risk, one should actually measure P losses. Assuming that analyzing agronomic P indices would reveal P losses is not enough. The last sentence of this paragraph is cryptic why it needs rephrasing and specifying the meaning.
115: Please tell how sampling was made, e.g. from how many subsamples a sample was composed of.
117-130: There seems to be repetition here, please rephrase.
133: Please indicate that WSP was calculated to mg/kg soil.
137: I assume 10% was the SOM content, not C?
145-146: “The basis for the assessment was the percentage share of the soil samples tested in individual reaction and abundance classes.” Please clarify. Do you mean that agronomic interpretation of STP is not based on STP and pH only, but the STP distribution also affects the class division? Maybe it’s the word abundance that makes the confusion – do you mean P concentration, or the number of soils in a STP class?
158-164: I’m not sure how these analyses were used, otherwise than for constructing the example Fig. 2. Maybe these were meant for Table 5 (that in its current seems unnecessary)?
179-184: Please describe more specifically the terms “medium”, “light”, “very light” soils. I suspect you’re meaning textural classes and naming the major soil separates and clay contents would be more easily understood (e.g., loamy soils with X-X% clay).
Could these data be combined with STP and WSP data to show if there are some obvious risk areas?
209-210: Be consistent in using abbreviations: PH2O -> WSP.
215: Please elaborate the conflict between agronomic and environmental interpretation after this paragraph. To be interesting to a reader, you should tell something about the sorption properties of the studied soils. Minimum would be extractable Al and Fe, e.g. by the oxalate method which is the most widely used. Otherwise there just are two opposite interpretations, of which one appears to be wrong but one cannot guess which. Are there some P fertilization studies done at the area (at least in line 262 P deficiencies are mentioned, please elaborate that)? Reference to such data would tremendously strengthen the paper. The recent paper of Fischer et al. (http://dx.doi.org/10.1016/j.scitotenv.2017.03.143) shows that the earlier German target STP was in practice always associated with high P loss risk (assessed according to Pöthig et al.). Because the target STP could be lowered in 2015, it is likely that there was a good safety margin associated with the old one. Is there a possibility that the interpretation in Poland could be similarly “on the safe side” as the past German agronomic interpretation? Or is P buffering of the Puck area soils so low that once P concentration in soil solution declines, there is no buffering by the adsorbed P that would replenish it?
When ending: “There was a high risk of P loss from these soils to water following the approach to determining this risk suggested by Pöthig et al. (2010).”, I wonder what is your own interpretation? If a soil contains about 25 mg WSP/kg, it would mean that P concentration in (1:50) water extract would be about 0.5 mg/L. To me it seems quite high, but how does it relate to dissolved P concentration in runoff waters? Can you compare with some Polish studies on runoff water quality?
220-222: Please re-read these sentences: “There was a relatively low correlation between the pH value of the soil and its P content determined by the Egner-Riehm method both in the layer up to 5 and 30 cm. However, this correlation indicates that by reducing the acidity of soils plant available P would increase.” I’m not sure that a poor correlation between pH and STP would indicate that plants could utilize more P if pH is elevated. Increasing pH of an acidic soil will make P more soluble, but justifying this conclusion by the preceding sentence is not sound reasoning.
227-228: Not sure the meaning of this sentence. Please define what you mean by “topsoil” (0-5 or 0-30 cm). Secondly, a correlation between P concentrations in these two sections does not prove that P was homogenously accumulated (I guess you mean dispersed), just that the concentrations rank in a relatively similar order. However, your supplemental data shows that there is no vertical stratification in STP concentrations.
228-230: Please clarify this sentence. Do you mean that it is equally good to use whole 0-30 cm section samples in environmental risk assessment? Is it a general recommendation, or valid for the study area only?
236-237: All abbreviations (PH2O_w1) should be opened when coming first time.
254-255: “… soils practically in the whole analysed area of agricultural lands were overly supplied with P…” Is it also possible that the studied soils’ P retention capacity is very low and therefore the environmental and agronomic interpretations disagree?
268: If “This treatment” refers to liming, note that liming in itself does not increase P resources, but the solubility of the reserve. But I totally agree that liming would be a first remedy for the soils that fall into the low pH categories.
270: Change “DPS-based analyses” to “WSP analysis” (WSP was analyzed, DPS was calculated).
276-278: Maybe it could be stated that P loss risk index is just one component and needs to be complemented with others that tell about the delivery from soil to water. Good references are found by searching “Lemunyon and Gilbert P index” and “P loss continuum”. The discussion section would in all cases benefit from more references.
Fig 1: Please indicate that the R2-value refers to the data of Pöhlig et al.
Fig. 4: Please add explanation that these are soil test P classes in the figure caption and replace “P content” with “Soil test P class”.
Table 1 caption: Use rather something like: “Soil test P concentrations according to the Polish agronomic interpretation…”, change “abundance” to “Soil test P”, “assessment” to “interpretation”, remove “ingredient”. There are unnecessary left parentheses markers in the table and > markers are on the wrong side of the range values.
Table 2 caption: Replace “thresholds” with “classes”.
Tables 3 and 4: Unnecessary row at the bottom (DPS).
Table 5: Should there be data of the study area in this table?
Supplemental: what is P-wod_w1? I have an impression that the Journal encourages publishing all data. I could not find anything about clay and SOM content. Also, a file with measured STP and WSP could be useful in development or calibration of agro-environmental models, and much appreciated by that community.

·

Basic reporting

This manuscript evaluates the performance of the universal DPS-index method for assessing the risk of phosphorus loss in surface runoff from agricultural land at the Baltic Commune of Puck. The manuscript is generally written is acceptable English, but there are many sentences where the writing style is not appropriate for scientific publication. The authors need to put significant effort into using the correct terms and avoid sentences like “(HCL solution is an aggressive extract, water – not)”.

The manuscript is very convoluted, particularly the “Results and Discussion” section, which is longwinded and difficult to read and extract the key findings and contributions. I suggest the authors to consider splitting this section into two sections, one for “Results” and one for “Discussion”. In the “Results” section, please simply describe and highlight the important results (plots and tables) and move the description of the case study to “Materials and Methods”. In the “Discussion” section, please examine your results in greater detail and compare your findings with the literature. This is the most important section of your paper and it should clearly identify the main contributions of your work (based on your results) to the scientific community – please build concise sentences and be direct to the point avoiding vague statements such as “, in environmental terms,”.

The tile does not reflect the actual contribution of the work presented. As said in the introduction “The aim of this work is to test the usefulness of the Pothig et al. (2010) method to assess the risk of P loss in…” I suggest the authors change the tile to something more informative, such as “Evaluating the performance of the universal DPS index method for assessing the risk of phosphorus loss in surface runoff from agricultural land: the case of the Baltic Commune of Puck”. This suggestion is a bit long, but this is just a suggestion for the authors to work on.

Finally, I also find that the scientific contribution of the paper is poor. The authors seem to have a substantial dataset in their hands – pH and P content for 61 farms. However, there is no map showing the location of these plots nor, for instance, an analysis of the spatial distribution of pH and P content - this is just an example of the type of work that could potentially be included to make the paper more interesting. Also, the measurements presented should be better described (e.g., when did these measurements took place) and should be related to hydrological processes - is there a temporal dependency worth to explore. Nothing is mentioned about the hydrology of the region, although it is well known that it can have a determinant effect on soil P concentrations. Also, to improve the quality of the manuscript, the authors could focus on improving the “universal” DSP-index method and propose a way forward – this would be a much stronger contribution.

For all the reasons above, I think that this paper requires profound improvements and, therefore, I recommend REJECTION, but I urge the authors to re-submit it after the issues identified have been addressed. I provide below many comments and suggestions that will help the authors to improve the quality of the paper.

Experimental design

The authors do not describe the experimental design in enough detail. I provide below examples of how this could be improved.

Validity of the findings

As said before, the "Results and Discussion" section is convoluted and needs major revisions. I provide below examples of how this could be improved.

Additional comments

GENERAL NOTES:
Line 39: I suggest changing slightly the portion “it was considered that the assessment based on the DPS index may be unreliable” to something more constructive or adding some more detail. For example: “…may be unsuitable for… (specify)”
Line 49: the citation (National…, 2016) is not properly handled. Probably use an acronym based on the what is in the reference listing, NWMB (standing for “National Water Management Board”).
Lines 51 to 58: In my opinion these sentences are unsuitable for a scientific paper. I would rephrase and remove for instance “made at the Conference of Ministers for the Environment of the HELCOM countries”
Line 63: Eutrophication parameters? What are these parameters? The authors should be more specific
Line 67: What “component”? Very vague; please make the sentence clearer
Lines 120-133: This is an article, not a report. In my opinion, the authors should adopt in-line numbering or simply integrate the information in “normal” sentences.
Line 136-150: This listing is unnecessary.
Line 155: I don’t think that “the results of laboratory analysis were also developed statistically”? Do you mean that statistical tests were performed on the lab results? Please clarify
Line 157: “Statistica 6”? The reader might not know what software this is. Please add a reference and maybe some details about what it does and what functionalities did you use for this particular study
Line 158-164: This whole paragraph is very confusing. Are you trying to say that you complemented your DSP-index analysis with a land use classification assessment? Please rephrase the sentence to be more direct to the point.
Line 167-169: This whole paragraph is not part of the results. It should be in the Materials & Methods section if not included there already. In this section (Results) focus only on the results.
Line 172-178: please remove this list and integrated the information in a proper sentence.
Line 179; This sentence is cumbersome. Please rephrase. And this is not really the results of this study. This is a general description of the study site and should me move to “Materials and Methods”
Line 195-196: what is P_ER and P_HCL? This seems to be the first time these are mentioned in the text so they need to be described.
Line 194-200: The whole paragraph is very cumbersome. E.g. “The available P content in the discussed layer of analysed soils…” and “P content in these in relation to 62.3% of plots was…”. Please revise profoundly.
Line 206: The authors need to improve the English significantly. What do you mean by “the P content in the shallower of the analyses”? are you referring to the topsoil?
Line 227: What do you mean by “The P content was most strongly correlated with 0-5 and 0-30 cm soil profiles,…”? This makes no sense. Do you mean correlated (or varied) with the soil depth?
Line 254: What do you mean by “in environmental terms”? Please just remove this type of statements all along as they don’t add anything and just make the text convoluted.

TECHNICAL NOTES AND SUGGESTIONS:
Line 21-22: suggest changing to: “… is a basic measure that should be included in action plans to counteract eutrophication in…”
Line 23: used either “, which” or “that”
Line 25: add “the” after “Based on”
Line 44-45: suggest changing to “… excessive river inflow of phosphorous (P).”
Line 45: add “estimated” after “… Baltic basin is”
Line 47: add “About” before “12.7 tons of…”
Line 61: replace “reasons” by “factors”
Line 63-64: replace “in” by “by” before “42%” AND add “by” before “66%”
Line 88: I suggest rephrasing this sentence to something like “This method is attractive because it is simpler and has the potential of being applicable to different soil types.” It is important to write this sentence carefully because it will create a certain expectation in the reader, and as said in the abstract “… the DPS index may be unreliable”. The message and objective of the paper should be clearly articulated throughout the paper.
Line 89-92: In this sentence “In the MONERIS model,… , it is particularly used…” it is unclear what “it” is referring to.
Line 93: I suggest replacing the work “usefulness” by “ability” AND replace “the Pothig et al method (2010)” by “the method proposed Pothig et al. (2010)”
Line 93: “method” should go after “(2010).
Line 94-95: Rephrase to something like: “This was accomplished by…”
Line 101: replace “The” by “This”
Line 101-107: Please rephrase this paragraph. Some suggestions: remove “by nature” AND replace “danger” by “risk” AND link better the different sentences
Line 108-114: this information is not relevant for the purpose of this paper. The purpose of this paper is to inform other scientists about the performance of a method, so please remove everything that isn’t important for this goal - just say that the study presented in this paper is part of a larger project/study and add a reference (and remove everything else)
Line 134: Replace by “… according to the following criteria: (1)…”
Line 203: add “the” before “P content”
Lines 187-188: no need to capitalized “manganese”, “iron” and “aluminium”

---

## Round 0.2 · Major Revisions

You have made some progress in addressing the reviewer comments, but there are still issues with your revised manuscript. In particular, the grammar still needs improvement. As I stated earlier, once you have made all the necessary revisions, I strongly suggest that you have the revised manuscript edited by a native English speaker. I cannot accept a revised manuscript that is not readable.

·

Basic reporting

The authors made a significant effort to improve the paper based on the comments provided by the two previous reviewers. However, I still think that the paper doesn't meet the minimum requirements for publication as grammar errors can still be found throughout the paper, as well as unclear sentences. Furthermore, I think that the paper is long-winded, which makes it very difficult to follow, particularly the Discussion section. I tried to provide suggestions on how to improve the overall quality, but it took me a long time because there are really too many English problems, as there were in the first version.
I also have concerns regarding the scientific contributions of this paper. It is said that "It should be recognized that the risk level of P losses in surface runoff from the analyzed plots determined by the DPS indicator is uncertain. This uncertainty is increased by the lack of linking the DPS index with other significant factors in P runoff losses such as type of crop and area inclination." However, the title of the paper is "Risk of phosphorus losses in surface runoff from agricultural land in the Baltic Commune of Puck in the light of assessment performed on the basis of DPS indicator”, which seems to be in conflict. If the method used is uncertain for the main objective of the paper, what is the contribution of the paper “risk of phosphorus loses?
The authors may still be able to publish the paper, but they’ll need to identify the really valuable contribution of their research work, which seems to be more on the applicability of the DPS indicator across different regions? One possibility could be to change the title to “Use of the DPS indicator for risk assessment of phosphorus losses in surface runoff: evaluation of the method and application to the Baltic Commune of Puck”, and the authors would need to make a significant effort to really identify the key contributions (take away messages) from this research, and maybe create sub-sections in the Discussion section with titles that summarise those key points and, then, only bring the critical information necessary to support that the conclusions in that sub-section inside “Discussion”.

Experimental design

No major comments here.

Validity of the findings

No major comments here.

---

## Round 0.3 · Minor Revisions

The reviewer has pointed out a few minor comments that when addressed will make your manuscript acceptable.

·

Basic reporting

The authors made a great effort to improve the overall quality of the manuscript. The manuscript is now well written and much more readable. I continue to disagree with the authors about the Discussion section, which I continue to think that it should be split into sub-sections. Presently, this section is 7 pages long, with the authors discussing different aspects of the results; one after the other. I provide again below some suggestions for ways in which that section could be split into sub-sections. This would also help to highlight the key messages as the authors go through their data, as well as make the whole section more digestible. This is my honest opinion, but I will leave the final decision on this matter to the authors. I provide below also a few minor suggestions, but I recommend "minor changes" to this manuscript as I think that the paper is now of good quality and ready for publication pending the authors addressing the few points listed below.

Experimental design

No comment.

Validity of the findings

No comment.

Additional comments

1) Line 100: remove “in the”
2) Line 231-233: I would elaborate on the spatial heterogeneity a bit more instead of saying only that it’s very heterogeneous. For example, it seems that the soil in the northwest areas tends to be more acidic and have a lower P content when compared to the southeast areas. Maybe you can elaborate on why that could be happening.
3) Discussion section: The authors do not agree with me in that the Discussion section should be divided into sub-sections. I respect the decision of the authors, although I utterly disagree. The discussion section is seven pages long; isn’t there a way to break this into smaller sub-section that would help to highlight the key messages? I can clearly identify a few:
3a) DPS index assessment suggests a high risk of P loss (lines 346-359)
3b) Fertilizer use to protect plant yields (lines 219-233)
3c) Comparing the DPS indicator to agronomic test methods
3d) ….

---

## Round 0.4 · accepted · Accept

Thank you for your considerable efforts in revising your manuscript and for your patience with the review process.